# Factors associated with neonatal pneumonia in India: protocol for a systematic review and planned meta-analysis

Sreekumaran Nair,[1] Leslie Edward Lewis,[2] Myron Anthony Godinho,[1] Shruti Murthy,[1] Theophilus Lakiang,[1] Bhumika T Venkatesh[1]

## ABSTRACT

**Introduction** India accounts for more neonatal deaths than any other country. There is a lack of consolidated evidence from India regarding the determining factors of pneumonia in neonates. This systematic review is aimed to consolidate and appraise the evidence on risk factors and determinants of pneumonia among neonates in India.

**Methods and analysis** This protocol is part of a project consisting of three reviews (two systematic reviews and one scoping review) and a qualitative study on neonatal pneumonia in India. English language observational studies which report risk factors and determinants of neonatal pneumonia in India will be eligible for inclusion. Electronic searching of nine databases, and hand searching will be done. Two authors will independently conduct screening (title, abstract and full-text stages), extract data and assess risk of bias. A meta-analysis is planned to be performed with random-effects model. A narrative synthesis will be used to summarise the characteristics and findings of the review, if a meta-analysis cannot be performed. If there are more than 10 studies, publication bias will be assessed. Sensitivity and subgroup analysis will performed based on data availability. The quality of our review will be assessed by using 'Assessing the Methodological quality of Systematic Reviews' and 'Grades of Recommendation, Assessment, Development and Evaluation'.

**Ethics and dissemination** The protocol of the entire project has been approved by the host institution's ethics body (Institutional Ethics Committee, Manipal University, Manipal, India), and the 'Health Ministry Screening Committee' under the Ministry of Health and Family Welfare, Government of India. The study findings will be disseminated among relevant stakeholders using knowledge dissemination workshops, policy briefs, publications, etc.

**PROSPERO registration number** CRD42016044019.

[1]Public Health Evidence South Asia (PHESA), Manipal University, Manipal, India
[2]Department of Pediatrics, Neonatology Unit, Kasturba Medical College, Manipal University, Manipal, India

**Correspondence to**
Dr Sreekumaran Nair; nsknairmanipal@gmail.com

## Strengths and limitations of this study

► This is the first systematic review and meta-analysis on risk factors of pneumonia among neonates in India.
► Use of a sensitive search strategy over nine databases, including regional databases and websites. Both published and grey literature will be searched.
► No time restriction will be applied for searches and inclusion.
► We will consider studies which reported on pneumonia among neonates independent of sepsis, and that published in English. We will capture the definitions of neonatal pneumonia as reported in included studies.

the global neonatal mortality rate (NMR) of 16.7 deaths per 1000 live births exceeded both postneonatal mortality (11.7 per 1000 live births) and mortality in the age group of 1–4 years (10.5 per 1000 live births). The South Asian NMR was estimated at 23.2 deaths per 1000 live births, with India's NMR of 21.8 deaths per 1000 live births being the second highest in the region, only exceeded by Pakistan (31.8 per 1000 live births).[1]

In India, neonatal mortality accounts for 57% of national under-5 deaths.[2]

Thirty-three per cent of neonatal mortality in India is due to infectious disease,[2] comprising pneumonia (16%), sepsis (15%) and diarrhoea (2%).[3] Pneumonia in neonates contributes significantly to mortality in infants, with over half of childhood pneumonia deaths occurring during the newborn period.[4]

Most studies have focused only on sepsis and pneumonia in children aged under 5 years. Neonates are considerably different from postneonatal children, both in their physiology and the environments

## INTRODUCTION

Globally, nearly 5 million children aged under 5 years died in 2016; regionally, South Asia contributed nearly a quarter (24.8%) of the global burden of under-5 mortality, with almost a million deaths occurring in India alone, the largest of any nation.[1] In 2016,

they are exposed to,[5] and are more affected by antenatal factors (eg, teenage pregnancy, poor spacing between successive births, etc).[6] Neonatal pneumonia is clinically categorised as 'early onset' or 'late onset'—each of which determined by factors in the intrauterine or the external environment, respectively.[7] Different arrays of pathogens produce different variants of neonatal pneumonia.[8] Early onset neonatal pneumonia is often due to pathogens aspirated by the neonate from the intrauterine environment, or from the birth canal during vaginal delivery. The risk is particularly high if the mother has chorioamnionitis—an infection of the intrauterine tissues.[9] Late-onset pneumonia is often caused by pathogens encountered in the postnatal environment, either in the community (community-associated pneumonia), or in the hospital (hospital-associated pneumonia). Neonates with breathing difficulties (commonly due to prematurity or hyaline membrane disease) often require mechanical ventilation through an endotracheal tube, which can serve as a route for introducing nosocomial pathogens, causing ventilator-associated pneumonia.[10] Pneumonia can be experienced by neonates who have dysphagia (difficulty swallowing) due to neuromuscular disorders, anomalies of the oropharyngeal anatomy (eg, cleft palate, tracheo-oesophageal fistulae) or who experience gastro-oesophagaeal reflux disease . Lipoid pneumonia is due to aspiration of oil, and is not unusual in India due to the regional sociocultural practice of bathing newborn children in oil or ghee, and cleaning their nose and mouth with the same.[11] In India, poverty, malnutrition, impaired access to healthcare, poor immunisation status, poor child-rearing practices, and indoor air pollution are common risk factors for pneumonia.[9 12]

Consolidated evidence on the factors responsible for the Indian burden of neonatal pneumonia is greatly lacking and urgently required. With this study, we propose to identify various risk factors and determinants associated with neonatal pneumonia, in order to better understand, identify and eliminate these risk factors and determinants, and thus reduce the disease burden of neonatal pneumonia in India.

This protocol is part of a project consisting of three reviews (two systematic reviews[13] and one scoping review[14]) and a qualitative study[15] on neonatal pneumonia in India. The other systematic review and planned meta-analysis will consolidate and appraise the evidence relating to the predictors of mortality due to neonatal pneumonia in India, and has been described elsewhere.[13] The scoping review will identify and summarise the evidence on the barriers to pneumonia case management and options for treatment.[14] In addition to the three reviews, a qualitative study will attempt to capture perceptions of stakeholders in the neonatal healthcare system.[15]

## METHODS AND ANALYSIS

This protocol is developed and reported based on the 'Meta-analysis of Observational Studies in Epidemiology'

(MOOSE) guidelines,[16] and 'Preferred Reporting Items for Systematic reviews and Meta-Analysis- Protocol' (PRISMA-P) guidelines.[17] This systematic review and meta-analysis will be conducted from August 2016 to January 2018.

### Criteria for considering studies for this review
### Types of studies

Inclusion criteria: observational studies, published in English, which report on risk factors or determinants of pneumonia among neonates in India will be considered eligible for inclusion. Studies can be either descriptive (eg, case series, cross-sectional design) or analytical in design (eg, case-control, cohort, analytical cross-sectional designs). Reports of secondary data analysis and fact sheets should provide risk factor estimates to be considered for inclusion in the review. Studies published in English language will be eligible for inclusion. It is unlikely that languages other than English would be the primary language in which the evidence reports would be published. Attempts to identify relevant material published in other languages will be made by contacting established networks and experts in the review topic to suggest any studies in local language on the topic.

Exclusion criteria: intervention studies, all reviews, meta-analyses, qualitative research, conference abstracts/papers, letters, editorials, commentaries, reports which do not provide risk factor estimates for neonatal pneumonia will be excluded. Since neonatal pneumonia is often considered a component of neonatal sepsis, we will consider such studies for inclusion if the study specifically reports on the pneumonia component explicitly.

### Type of participants: neonates in India

Outcome of the review include risk factors and determinants of neonatal pneumonia. Definitions of 'risk factor' and 'determinant' as reported by the included studies will be captured in our review. Examples of operational definitions of these key terms are discussed below.

### Risk factor

'A risk factor is any attribute, characteristic or exposure of an individual that increases the likelihood of developing a disease or injury (or other negative/undesirable outcomes)'.[18]

### Health determinant

'Underlying personal/social/economic/environmental characteristics which ultimately shape the health of individuals and communities. They are "upstream factors" or, the causes of ill health'.[19]

Table 1 outlines possible (not exhaustive) risk factors for pneumonia among neonates, which have been adapted through consultation with subject experts and from existing literature.[20–23] This list will be modified based on the findings from this systematic review.

### Search methods for identification of studies

An exhaustive list of search terms will be used to develop a sensitive search strategy for final searches. There will be

**Table 1** Possible risk factors of neonatal pneumonia

| Category | Factors |
|---|---|
| Patient-related | Age<br>Gender<br>Low birth weight<br>Small-for-age<br>Prematurity<br>Birth order<br>APGAR score at birth<br>Concomitant conditions/comorbid illnesses (eg, congenital heart disease, asthma, haemoglobin, cyanosis, hypothermia)<br>Congenital malformations<br>Feeding practices<br>Others |
| Parent-related | Sociodemographics<br>Economic<br>Cultural practices in the context<br>Medical history<br>Vaccination<br>Healthcare-seeking knowledge, attitude, behaviour<br>Family planning variables<br>Others |
| Maternal and pregnancy- related | Gestational age<br>Intrauterine growth restriction<br>Conditions/diseases in pregnancy<br>Urinary tract infections<br>Mode and place of delivery<br>Skilled birth attendance<br>Foul-smelling liquor<br>Prolonged rupture of membranes<br>Prolonged labour<br>Intrapartum fever<br>Others |
| Environment-related | Place of residence (eg, urban, rural, periurban)<br>Housing<br>Overcrowding<br>Water, sanitation and hygiene<br>Indoor air pollution (including cooking smoke, mosquito coils/repellents, secondhand smoke exposure)<br>Pets/livestock<br>Rainfall<br>Outdoor air pollution<br>Seasonality<br>Others |
| Health system - related | |
| Iatrogenic | |

**Table 2** Search strategy for PubMed

**Strategy: #1 AND #2 AND #3**

| | |
|---|---|
| #1 | (((Neonate* OR childhood OR neonatal* OR newborn* OR 'young infant' OR child OR pediatric* OR paediatric* OR 'neonatal period' OR infant* OR 'newborn infant')) |
| #2 | ((((((((((((((((((((((((((Pneumonia*) OR Pneumon*) OR 'community acquired pneumonia') OR 'congenital pneumonia') OR 'hospital acquired pneumonia') OR 'nosocomial pneumonia') OR 'ventilator associated pneumonia') OR 'early onset pneumonia') OR 'late onset pneumonia') OR 'infective pneumonia') OR 'infectious pneumonia') OR 'meconium aspiration syndrome') OR 'meconium aspiration') OR 'lipoid pneumonia') OR sepsis*) OR 'acute respiratory infections') OR 'early onset sepsis') OR 'chemical pneumonia') OR 'aspiration pneumonia') OR 'late onset sepsis') OR infection*) OR 'nosocomial infection') OR 'early onset infection') OR 'late onset infection') OR 'acute lower respiratory infection') OR 'hospital acquired infection') OR 'congenital infection') OR 'viral pneumonia') OR 'gastroesophageal reflux disease') OR 'cystic fibrosis') |
| #3 | (((('Risk factor' OR determinant* OR risk* OR predictor* OR 'relative risk' OR 'odds ratio' OR 'attributable risk' OR 'population attributable fraction')))) |

Geographical filter: India.
Language filter: English.

no restrictions on the time period for database searches, as studies published at any time are eligible for inclusion.

Electronic searches will be done on nine databases: PubMed, Ovid MEDLINE, Web of Science, CINAHL, EMBASE, SCOPUS, IMSEAR, ProQuest and IndMED. Handsearching will be performed for those journal volumes/issues that are not available in electronic searches and for conference proceedings to review the references and contact the authors for full texts of identified literature. Dissertations/thesis will be identified on Shodhganga (INFLIBNET), and reports/fact sheets will be retrieved by searching national websites. Snowballing will be done to identify relevant studies from reference lists of included studies and systematic reviews. However, such systematic reviews will not be eligible for inclusion in our review. Moreover, subject experts and authors of identified studies/organisations will be contacted to procure relevant studies for inclusion in the review. Table 2 contains an example of our search strategy.

### Data collection and management

Endnote (V.x7) will be used to manage the search results and perform screening. The data extraction will be performed on Microsoft Excel 2007, and STATA (V.13) will be used for statistical analysis.

### Selection of studies

Screening of search results for eligible studies will be performed by two authors (MG and SM) independently in title, abstract and full-text stages. During the title screening stage, titles approved by either of the authors will proceed to abstract screening. At the abstract screening stage, the abstracts that are approved by either of the authors will proceed to full-text screening. Abstracts rejected by both authors will be excluded. During full-text screening, those records that are approved by both the authors will

be included in our review. Disagreements will be resolved through consensus with an arbitrator (TL) and/or senior review authors (NSN and LEL). A PRISMA chart will be used to illustrate the study selection process.[24]

### Data extraction
Data extraction will be performed on pilot-tested spreadsheet by two authors (MG and SM) independently. Disagreements between authors will be resolved by arbitrated consensus. The data extraction form has been provided in the online supplementary file, and was developed in consultation with subject and methodological experts. Pilot-testing of the data extraction form was performed on different study designs to ensure that it is complete and adequate enough for various study designs. Data will be captured for study details, methodological characteristics, risk factors and determinants and their measure and strength of association and other information (eg, funding, limitations, conflict of interest, etc).

### Dealing with missing data
The authors of studies with missing data or methodological information will be contacted by email to obtain the required data/information. If missing data cannot be retrieved despite this effort, the particular study will be excluded from the meta-analysis, and a narrative summary will discuss the findings on risk factors/determinants from the study.

### Risk of bias assessment
Risk of bias assessment will be conducted two authors (SM and TL) independently, at the study level. Checklists will be based on the design of the study as follows:
► Case-control study: Newcastle-Ottawa Scale (NOS)[25];
► Cohort study: NOS[25];
► Cross-sectional study: an adapted version of NOS[26];
► Case series: Institute of Health Economics criteria[27].

Disagreements will be resolved through discussion and consensus in the presence of a third (MG) or senior author(s) (NSN and LEL).

### Data analysis
A meta-analysis will be performed using a random-effects model. The $I^2$ statistic will be calculated and reported. Categorical data will be summarised using OR and/or relative risk (RR). Continuous data will be summarised using the standardised mean difference. The summary measures will be pooled together based on the study design. A forest plot will be generated and pooled estimates will be reported with 95% CIs.[28]

Based on the availability of data, a subgroup analysis will be conducted according, but not limited, to study design, study setting, the type of neonatal pneumonia and the timing of onset of pneumonia among neonates. A sensitivity analysis will be performed to determine the robustness of the findings, by removing one study at a time and noting the change in pooled estimate. Reporting bias will be assessed if there are more than 10 studies that are included in our review. A funnel plot will be generated and

Egger's test performed to assess the degree of asymmetry. Based on the data availability, attempts will be made to perform a meta-regression to assess covariates' effect on the pooled estimate, and adjust for this effect. The characteristics and findings of the review will be presented in the form of tables and a narrative summary. Additionally, the narrative summary will discuss the impact of study quality and limitations of included studies on the findings.

*Synthesis of project findings*: the analysis plan for the collation of the findings from the three reviews and the qualitative study will be described elsewhere.

### Quality assessment of the systematic review and meta-analysis
'Assessing the Methodological quality of Systematic Reviews' criteria will be used to evaluate the systematic review's methodological rigour.[29] The 'Grading of Recommendations Assessment, Development and Evaluation' approach will be used to assess the quality of evidence generated, and a summary of findings table will be generated and reported using GRADEPro.

### Ethics and dissemination
#### Ethics
The protocol of the entire project consisting of three reviews, a qualitative study and data triangulation has been approved by the host institution's ethics body, (Institutional Ethics Committee at Manipal University, Manipal, India), and the 'Health Ministry Screening Committee' under the Ministry of Health and Family Welfare, Government of India.

#### Dissemination
The study findings will be disseminated among relevant stakeholders through knowledge dissemination workshops, publications, plain language summaries, policy brief(s), etc.

#### Reporting of the systematic review and meta-analysis
The PRISMA[16] and MOOSE guidelines[24] will be used to report our review.

**Acknowledgements** The authors would like to thank the following persons for their guidance and support throughout our protocol development process: Dr Manoj Das, Director Projects, The INCLEN Trust International, New Delhi; Dr Anju Sinha, Deputy Director General, Scientist 'E', Division of Child Health, Indian Council of Medical Research, New Delhi; Dr KK Diwakar, Professor and Head, Department of Neonatology, Associate Dean, Malankara Orthodox Syrian Church Medical College, Kerala; Mrs Ratheebhai V, Senior Librarian and Information Scientist, at Manipal School at Communication, Manipal University, Manipal; Dr Ravinder M Pandey, Professor and Head, Department of Biostatistics, All India Institute of Medical Sciences, New Delhi; Dr B Shantharam Baliga, Professor, Department of Paediatrics, Kasturba Medical College, Mangalore, Karnataka; Dr Shirish Darak, Senior researcher, PRAYAS, Pune, Maharashtra; Dr Unnikrishnan B, Associate Dean and Professor, Department of Community Medicine, Kasturba Medical College, Mangalore. The authors would like to thank Public Health Evidence South Asia (PHESA), Manipal University, for providing institutional and infrastructural support. The authors would also like to thank The INCLEN Trust International, New Delhi, and The Bill and Melinda Gates Foundation for their financial support.

**Contributors** NSN is the guarantor of the review. NSN, BTV and LEL conceived the research idea and reviewed the manuscript. NSN and LEL provided overall technical guidance. In addition, LEL assisted in developing search terms. MG, SM and TL

designed the protocol, drafted the manuscript and developed and pilot tested the search strategies and data extraction form.

**Funding** This project is supported by a grant from Bill & Melinda Gates Foundation (grant OPP1084307) to The INCLEN Trust International and s ubgrant to Manipal University (subgrant INC2015GNT004).

**Disclaimer** The views expressed through this project do not necessarily represent the views of Bill and Melinda Gates Foundation or The INCLEN Trust International or Manipal University.

**Competing interests** All authors have completed the ICMJE uniform disclosure form at www.icmje.org/coi_disclosure.pdf and declare: all a uthors had financial support (grants) from Bill and Melinda Gates Foundation (grant OPP1084307) to The INCLEN Trust International and subgrant to Manipal University (subgrant INC2015GNT004), during the conduct of the study and for the submitted work; no financial relationships with any organisations that might have an interest in the submitted work in the previous 3 years; no other relationships or activities that could appear to have influenced the submitted work.

**Provenance and peer review** Not commissioned; externally peer reviewed.

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
