## [Reviewer comments · BMJ Open]

ARTICLE DETAILS

TITLE (PROVISIONAL)	Factors associated with neonatal pneumonia in India: protocol for a systematic review and planned meta-analysis
AUTHORS	Nair, N.; Lewis, Leslie; Godinho, Myron; Murthy, Shruti; Lakiang, Theophilus; Venkatesh, Bhumika

VERSION 1 – REVIEW

REVIEWER	Dr Sadia Riaz Children Hospital, Pims Islamabad, Pakistan
REVIEW RETURNED	30-Aug-2017

GENERAL COMMENTS	It needs a major revision as the paper does not clearly address the study question If written clearly it's a good topic
--

REVIEWER	Diane Gray University of Cape Town Cape Town South Africa
REVIEW RETURNED	26-Sep-2017

GENERAL COMMENTS	This protocol describes a planned systematic review and meta-analysis of studies assessing risk factors for and determinants of neonatal pneumonia in India. It is clearly and rigorously written, following recommended methods. I have a few comments that the authors may want to consider at this protocol stage: 1. I do not agree with the authors that ethics approval is definitely not required for this systematic review. They state that this is because only secondary data already in the public domain is being used and that it does not involve human participants. But all these studies will be dealing with human participants and although ethics approval is not always clearly necessary in a systematic review, it is good not to assume waiver applies in all cases, especially where research deals with vulnerable human participants, in this case infants. I would have thought it appropriate for the authors to have at least presented this proposal to their local IRB who could waiver approval if they thought appropriate.
--

	Particular concerns that might want to be considered is whether the outcomes of the review are similar to the original studies for which consent was given; ensure there is no potential harm for participants from undertaking review (usually unlikely in anonymised data such as this) and that it holds benefit for them or their community; whether contacting authors for clarification may include unpublished details/data that may not have been included in original consent; how to manage included studies with ethical concerns. It is very likely that approval would be given in a case like this and would avoid any obstacles to publication of this data at a later stage - even if the IRB in question has only given an official waiver. 2. Exclusion of studies: the review will be limited to articles published in English. Is there likely to be relevant material published in other languages? It would be prudent to include these studies if they exist and fulfil other inclusion criteria. 3. Have the authors considered how they will deal with included studies that may have ethical concerns? Will they be excluded or included with discussion around impact of concern. This should be included in methodology. 4. The study is part of a planned "trilogy of reviews on neonatal pneumonia in India". Please could the authors describe the three reviews in the methods and analysis so that it is clear how this review fits into the plan without overlap or contradiction. 5. Please include the data extraction form. According to the methods this has been developed and piloted. It would clarify the details of outcomes and how the analysis plans fit if data extraction form could be reviewed. 6. The authors explain to some extent the planned outcomes -but could they please clarify in the text the primary outcome measures e.g. which risk factors and determinants will be collected and analysed. Would be helpful to specify these in the methods section. At this stage the protocol does not give a clear idea of which risk factors and determinants are expected to be relevant based on global and Indian-local data.
--	---

REVIEWER	Dr. GA Tramper-Stranders Pediatrician/clinical researcher Franciscus Gasthuis & Vlietland Rotterdam the Netherlands.
REVIEW RETURNED	04-Oct-2017

GENERAL COMMENTS	The manuscript is clearly written and is a good introduction of a systematic review according to the PRISMA guidelines. I am very curious about the outcomes. My main issue with this manuscript is the fact that it is a study protocol of a review. Although study protocols are accepted by BMJ Open, these study protocols are usually related to starting or ongoing clinical trials to increase transparency and the methodology. I would advice the authors to register the systematic review in PROSPERO and submit the manuscript together with the results of the review and meta-analysis,
---

	also since the conduction of the review is scheduled Aug 2016-Oct 2017, which is due now (therefore my suggestion of a major revision, although the protocol is well-written). Moreover, September 6, 2017; a review protocol belonging to the same series (trilogy) of this group (factors associated with mortality due to neonatal pneumonia in India) was published in BMJ Open. Although it is not a redundant manuscript per se because the objective is different, the texts of this recently published protocol and the submitted protocol are largely identical. Also, Sep 15 the protocol 'Treatment options and barriers to case management of neonatal pneumonia in India: a protocol for a scoping review' was published. I would have suggested to take the 3 reviews together for 1 protocol manuscript. With respect to the specific questions: Although the outcomes are defined as risk factors and determinants of neonatal pneumonia, in my opinion these could be more clearly defined. The authors state that the definitions will be captured in the review.
--	---

VERSION 1 – AUTHOR RESPONSE

Reviewer: 1

Reviewer Name: Dr Sadia Riaz

Institution and Country: Children Hospital, Pims Islamabad, Pakistan

Competing Interests: No

Comment: It needs a major revision as the paper does not clearly address the study question
If written clearly it's a good topic

Response: We have now revised the manuscript according to the comments provided by the editor and other reviewers, and anticipate that these revisions will meet the reviewers' requirements.

Reviewer: 2

Reviewer Name: Diane Gray

Institution and Country: University of Cape Town, Cape Town, South Africa

Competing Interests: No competing interests

This protocol describes a planned systematic review and meta-analysis of studies assessing risk factors for and determinants of neonatal pneumonia in India. It is clearly and rigorously written, following recommended methods.

I have a few comments that the authors may want to consider at this protocol stage:

1. I do not agree with the authors that ethics approval is definitely not required for this systematic review. They state that this is because only secondary data already in the public domain is being used and that it does not involve human participants. But all these studies will be dealing with human participants and although ethics approval is not always clearly necessary in a systematic review, it is good not to assume waiver applies in all cases, especially where research deals with vulnerable human participants, in this case infants. I would have thought it appropriate for the authors to have at least presented this proposal to their local IRB who could waiver approval if they thought appropriate.

Particular concerns that might want to be considered is whether the outcomes of the review are similar to the original studies for which consent was given; ensure there is no potential harm for participants from undertaking review (usually unlikely in anonymised data such as this) and that it holds benefit for them or their community; whether contacting authors for clarification may include unpublished details/data that may not have been included in original consent; how to manage included studies with ethical concerns. It is very likely that approval would be given in a case like this and would avoid any obstacles to publication of this data at a later stage - even if the IRB in question has only given an official waiver.

Response: The need for ethics approval for systematic reviews has been debated and most journals do not require ethics committee review/ approval for systematic reviews. The BMJ Open requires the submission of the PRISMA-P checklist for review protocols. The PRISMA-P checklist “does not contain item(s) for the ethical assessment of the studies included”, which has also been reflected in the article titled, “Ethics in systematic reviews” by Noel-Vergnes et al., 2010. Having said that, the protocol for the entire project consisting of three reviews, a qualitative study and data triangulation was presented and approved by relevant authorities in India. We have provided a statement with details of the same as follows:

“The protocol of the entire project consisting of three reviews, a qualitative study and data triangulation has been approved by the host institution’s ethics body, (Institutional Ethics Committee at Manipal University in Manipal, India), and the ‘Health Ministry Screening Committee’ (HMSC) under the Ministry of Health and Family Welfare, Government of India.”

2. Exclusion of studies: the review will be limited to articles published in English. Is there likely to be relevant material published in other languages? It would be prudent to include these studies if they exist and fulfil other inclusion criteria.

Response: We acknowledge that this could be a potential limitation of our review. However, on consultation with experts on searching regional databases, it was found to be unlikely that regional evidence reports would be published in languages other than English. Attempts to identify relevant material published in other languages will be made by contacting established networks and experts in the review topic to suggest any studies in local language on the topic. The same has been mentioned in the manuscript.

3. Have the authors considered how they will deal with included studies that may have ethical concerns? Will they be excluded or included with discussion around impact of concern. This should be included in methodology.

Response:

We will not exclude studies based on ethical issues of the included studies. However, we will perform an assessment of the reporting of the included studies and provide a discussion paragraph on the same to highlight the compliance to the reporting guidelines relevant to the included study design. We will perform this as one separate exercise for all the included studies of the trilogy of reviews neonatal pneumonia in India in this project, and will plan this as a separate manuscript.

Our approach to this is based on recommendations as found in the article “Ethics in systematic reviews” by Noel-Vergnes et al., 2010, which states: “that authors of systematic reviews should guarantee a minimum of ethical assessment and at least provide a brief discursive report of the ethical assessment of original studies. We suggest that, once the ethical characteristics have been collected, they can be summed up in a short discussion paragraph or in the form of a descriptive table, depending on the nature or the number of ethical irregularities. It is up to the author of the systematic review, depending on his field of research, to exclude certain studies as unacceptable from an ethical point of view and/or to analyze sub-groups according to different ethical parameters.”

4. The study is part of a planned "trilogy of reviews on neonatal pneumonia in India". Please could the authors describe the three reviews in the methods and analysis so that it is clear how this review fits into the plan without overlap or contradiction.

Response: The related protocols (which includes the trilogy and the qualitative study) have been mentioned in the introduction (as requested by the editor) and analysis sections of the manuscript. We plan to publish a separate document explaining the process and methods for the triangulation of findings from the three reviews and the qualitative study.

The revised sections now read as below:

Introduction:

"This protocol aims to describe the methods that will be used to conduct a systematic of the evidence on risk factors and determinants of with neonatal pneumonia in the Indian context. With this study, we propose to identify various risk factors and determinants associated with neonatal pneumonia, in order to better understand, identify and eliminate these risk factors and determinants, and thus reduce the disease burden of neonatal pneumonia in India.

This protocol is part of a project consisting of three reviews (two systematic reviews¹³ and one scoping review¹⁴) and a qualitative study¹⁵ on neonatal pneumonia in India. The other systematic review and planned meta-analysis will consolidate and appraise the evidence relating to the predictors of mortality due to neonatal pneumonia in India, and has been described elsewhere.¹³ The scoping review will identify and summarize the evidence on the barriers to pneumonia case management and options for treatment.¹⁴ In addition to the three reviews, a qualitative study will attempt to capture perceptions of stakeholders in the neonatal healthcare system.¹⁵"

Data analysis:

"Synthesis of project findings: The analysis plan for the collation of the findings from the three reviews and the qualitative study will be described elsewhere."

5. Please include the data extraction form. According to the methods this has been developed and piloted. It would clarify the details of outcomes and how the analysis plans fit if data extraction form could be reviewed.

Response: The data extraction form has now been provided.

6. The authors explain to some extent the planned outcomes -but could they please clarify in the text the primary outcome measures e.g. which risk factors and determinants will be collected and analysed. Would be helpful to specify these in the methods section. At this stage the protocol does not give a clear idea of which risk factors and determinants are expected to be relevant based on global and Indian-local data.

Response: Thank you for this feedback. We have incorporated the following additions in the methods section when discussing the 'Outcomes of the Review':

"Table 1 outlines possible (not exhaustive) risk factors for pneumonia among neonates, which have been adapted through consultation with subject experts and from existing literature.^{20,21,22,23} This list will be modified based on the findings from this systematic review.

Table 1: Possible risk factors of neonatal pneumonia"

(see revised manuscript for Table 2)

Reviewer: 3

Reviewer Name: Dr. GA Tramper-Stranders

Institution and Country: Pediatrician/clinical researcher, Franciscus Gasthuis & Vlietland, Rotterdam, the Netherlands.

Competing Interests: None declared.

Comment: The manuscript is clearly written and is a good introduction of a systematic review according to the PRISMA guidelines. I am very curious about the outcomes. My main issue with this manuscript is the fact that it is a study protocol of a review. Although study protocols are accepted by BMJ Open, these study protocols are usually related to starting or ongoing clinical trials to increase transparency and the methodology.

Response: BMJ Open is one of the few journals which accepts study protocols of systematic reviews. A quick search on BMJ Open for published review protocols turns up at least 27 study protocols of reviews for between September to October 2017. Publishing review protocols in BMJ Open is therefore not unusual or the exception.

As demonstrated by the growing number of published review protocols, there is an increasing recognition of the importance of publishing protocols of systematic reviews as it establishes a system of accountability for conducting and publishing the final review. Moreover, protocols of reviews on neonatal health are rare, especially in the Indian context, and it is trend that we are attempting to address.

Comment: I would advice the authors to register the systematic review in PROSPERO and submit the manuscript together with the results of the review and meta-analysis, also since the conduction of the review is scheduled Aug 2016-Oct 2017, which is due now (therefore my suggestion of a major revision, although the protocol is well-written).

Response: The protocol has already been registered in PROSPERO in October 2016, and the registration number was provided in the previous submitted version of the manuscript as "PROSPERO 2016:CRD42016044019". The PROSPERO citation has been provided for your reference below:

N. Sreekumaran Nair, Leslie Lewis, Myron Godinho, Shruti Murthy, Theophilus Lakiang, Bhumika T. Venkatesh. Risk factors for neonatal pneumonia in India: a systematic review and meta-analysis. PROSPERO 2016:CRD42016044019. Available from

http://www.crd.york.ac.uk/PROSPERO/display_record.asp?ID=CRD42016044019

Regarding the time period for the review, we have modified it to reflect as October 2016- January 2018, as we are awaiting the publication of this particular manuscript. We are aware that protocols are acceptable for publication until the data extraction stage of the review. Hence we have put the review procedures post-study selection on hold, until the publication of this manuscript.

Comment: Moreover, September 6, 2017; a review protocol belonging to the same series (trilogy) of this group (factors associated with mortality due to neonatal pneumonia in India) was published in BMJ Open. Although it is not a redundant manuscript per se because the objective is different, the texts of this recently published protocol and the submitted protocol are largely identical. Also, Sep 15 the protocol 'Treatment options and barriers to case management of neonatal pneumonia in India: a protocol for a scoping review' was published. I would have suggested to take the 3 reviews together for 1 protocol manuscript.

Response: We submitted all the three protocols together/ in close proximity. However, due to some issues with the third party submission system, the processing of this protocol was delayed. The objectives, as the reviewer rightly noticed, of the three reviews are different and have been formulated to have no overlaps whatsoever. Further, it has been recommended by our advisory members and project coordinating authorities that having too many outcomes in a single protocol, especially when they are unrelated within the domain (e.g. risk factors vs barriers to case management) is not advised. Additionally, while two are systematic reviews and planned meta-analysis, the third is a scoping review. Considering the above, we considered conducting three separate reviews to be the most-effective strategy.

Regarding the text of the manuscript, we have meticulously attempted to word the texts of the three manuscripts in a way to prevent them being largely identical to each other. Having said that, some methods sections (e.g. search restrictions, data collection and analysis) are standardized for systematic reviews, and cannot be worded very differently.

Comment: With respect to the specific questions:

Although the outcomes are defined as risk factors and determinants of neonatal pneumonia, in my opinion these could be more clearly defined. The authors state that the definitions will be captured in the review

Response: While it would be desirable to provide a definition in the protocol, our interest was to capture the definitions from the review. This was decided since there are differing definitions for neonatal pneumonia in different contexts, and the lack of a standardized definition. We are interested to review the evidence to highlight this issue/ or the lack thereof, and highlight the evidence to the authorities in charge of the neonatal health care system in India. Additionally, we will be using these findings to triangulate with that found from the qualitative study.

Additionally, we have included the following additional text explaining our operational definitions for better clarity:

“Examples of operational definitions of these key terms are as follows:

Risk factor: “A risk factor is any attribute, characteristic or exposure of an individual that increases the likelihood of developing a disease or injury (or other negative/undesirable outcomes)”.¹⁸

Health determinant: “Underlying personal/social/economic/environmental characteristics which ultimately shape the health of individuals and communities. They are ‘upstream factors’ or, the causes of the causes of ill health”.¹⁹”

VERSION 2 – REVIEW

REVIEWER	Diane Gray University of Cape Town, South Africa
REVIEW RETURNED	13-Nov-2017
GENERAL COMMENTS	The authors have adequately addressed my original comments.